# The Robust Study of Deep Learning Recursive Neural Network for Predicting of Turbidity of Water

**Shiuan Wan [1], Mei-Ling Yeh [2,\*], Hong-Lin Ma [3] and Tein-Yin Chou [4]**

[1] Department of Information Technology, Ling Tung University, Taichung 40851, Taiwan; shiuan123@teamail.ltu.edu.tw
[2] GIS Research Center, Director, Feng Chia University, Taichung 40724, Taiwan
[3] GIS Research Center, Feng Chia University, Taichung 40724, Taiwan; elmer@gis.tw
[4] GIS Research Center, Dean, Feng Chia University, Taichung 40724, Taiwan; jimmy@gis.tw
\* Correspondence: milly@gis.tw

**Abstract:** Water treatment is an important process, as it improves water quality and makes it better for any end use, whether it be drinking, industrial use, irrigation, water recreation, or any other kind of use. Turbidity is one of the fundamental measurements of the clarity of water in water treatment. Specifically, this component is an optical feature of the amount of light on scatter particles when light is shined on a water sample. It is crucial in water reservoirs to provide clean water, which is difficult to manage and predict. Hence, this study focuses on the use of robust deep learning models to analyze time-series data in order to predict the water quality of turbidity in a reservoir area. Deep learning models may become an alternative solution in predicting water quality because of their accuracy. This study is divided into two parts: (a) the first part uses the optical bands of blue (B), green (G), red (R), and infrared (IR) to build a regression function to monitor turbidity in water, and (b) the second part uses a hybrid model to analyze time-series turbidity data with the recursive neural network (RNN2) model. The selected models' accuracies are compared based on the accuracy using the input data, forecasting level, and training time. The analysis shows that these models have their strengths and limitations under different analyzed conditions. Generally, RNN2 shows the performance regarding the root-mean-square error (*RMSE*) evaluation metric. The most significant finding is that the RNN2 model is suitable for the accurate prediction of water quality. The *RMSE* is used to facilitate a comparison of the accuracy of the sampling data. In the training model, the training data have an *RMSE* of 20.89, and the testing data have an *RMSE* of 30.11. The predicted R-squared values in the RNN2 model are 0.993 (training data) and 0.941 (testing data).

**Keywords:** water quality; deep learning; recursive neural network

## 1. Introduction

Mountain areas can store large amounts of water in various ways for human beings. In Taiwan, watershed areas can be affected by different man-made developments; abuse; or natural factors, such as typhoons, heavy rains, and other soil and sand disasters. For instance, landslides are often directly or indirectly influenced by the turbidity of the water source and can cause reservoir siltation, seriously threatening water quality. Moreover, the variations in rainfall and soil/rock characteristics in reservoirs are also dominant factors of water quality [1–3]. A smart image management platform is required for water resources, and it would also help to strengthen water conservation and land use management. Past studies that have monitored the environment can be divided based on the two types of systems used: a monitoring system or a decision support system. Most of these systems are built based on the consideration of the factors of the water body and the land use surrounding the reservoirs [4]. Although water quality comprises a very broad area, such as hydro-chemical and physicochemical properties, this study only focuses on the turbidity

value of water, but other influencing factors of water quality that can also be predicted by our proposed method.

The Water Resources Department (WRD) in Taiwan effectively and constantly monitors the variation in water quality. Most turbidity problems are caused by landslides after heavy rain or illegal land use in the watershed area. However, developing intelligent image recognition technology is well accepted for water conservation and land management in the watershed area. Accordingly, the goal of this study is to construct a prediction-based image system to gain a better understanding of reservoir areas and water quality protection areas. This type of system safeguards the ecological conservation of water collection areas to ensure the safety and cleanliness of national drinking water, which hopefully ensures the sustainable use of water resources [5]. Regarding the effective management and control of image information, it can easily determine the location of actual violations and understand the results of image interpretation. Hence, the Water Resources Department (WRD) built the "Water Resource Conservation Smart Image Management Platform" in 2018. It is more effective in improving the convenience of the overall operation of image management, strengthening the functionality of the platform. The spatial management platform that we are developing will use the advantages of geographic information systems in relational analysis [6,7]. This imaging system should contain two major functions: (a) the ability to transfer image data to compute turbidity and (b) the ability to build a predictive water system to monitor turbidity in a reservoir. Specifically, the imaging system has an interface with related demand areas, it is connected to electronic maps and uses aerial images as base maps, and it constructs spatial displays to provide different images and real-time information references from the Water Resources Department.

However, the variation in water quality is hard to predict, and it may also be influenced by climate factors [8]. Turbidity in water is a major factor, and it is purified by water treatment plants in downstream in the water system. Thus, if an imaging system providing real-time data can transfer information to govern water quality, it will be of great help in understanding the current situation of the water [4]. In addition, if a proper predictive water system can be constructed, it will offer a better understanding of the near feature of turbidity in water. To achieve the prediction of water variation, a time-series neural network is adopted in this study.

As previously mentioned, the observations and predictions of water components (such as turbidity) seem to be a crucial problem for water treatment plants and should be a precaution when handling the current stage and next stage of the water body. Accordingly, in the current stage, one can consider using regression analysis to solve the problem. However, prediction involves a large amount of know-how in data analysis. Deep learning methods show promise for time-series predictions, for example, the automatic learning of temporal dependence and the automatic handling of temporal structures, such as trends and seasonality. With clear explanations, standard Python libraries (Keras and TensorFlow 2) have many useful analysis tools that can be used to carry out solutions for many fields, such as the prediction of PM 2.5 [9]. Furthermore, long short-term memory (LSTM) is another artificial recurrent neural network (RNN) architecture used in the field of deep learning. Currently, LSTM networks are applied to classify, process, and make predictions based on time-series data, since there can be lags of unknown duration between important events in an analysis of time-series data [10–12]. Therefore, using these tools to govern step-wise data and to train a neural network could be a possible way to resolve turbidity in water. The water quality of reservoirs, such as Qingtan Weir, is affected by heavy rains. This study plans to develop a boosting neural network to calculate the huge changes in turbidity under sudden conditions. The measured turbidity value is established in each sub-routine to correct the prediction of the drastic change in water turbidity of RNN2.

The two main types of analytical neural networks in the time series are (a) recurrent neural network (RNN1) and (b) recursive neural network (RNN2) [10–12]. The connections between the neurons of the time recurrent neural network form a directed graph, while the structural recurrent neural network uses a similar neural network structure to recursively

construct a more complex deep network. Time recursive neural networks can describe dynamic time behavior, and they are unlike feedforward neural networks, which accept a more specific structure input. RNN2 cyclically transmits the state in its network. That is, it can accept a wider range of time-series structures. For example, handwriting recognition was the first research result to successfully use RNN2. In general, the accuracy of the classifier can usually be improved by averaging the decisions of the classifier set. In general, when the various classifiers are different and accurate, a greater improvement can be expected. It was decided to obtain the results by adopting a basic learning algorithm and using it in multiple time domains on different training sets [13].

To achieve the goal of this study, we used the image data of the turbidity of Qingtan Weir, Taiwan. The sensor under the water attained the real value of the turbidity. Moreover, a UAV took an image to acquire the band value of the water body at the sensor. The regression model was built, and the time-series data were used to predict the turbidity change in Qingtan Weir after heavy rains or after soil condition changes through the RNN2 model. The paper is structured as follows: (1) Introduction, (2) Materials and Methods, (3) Methods, and (4) Discussion of the Results. The last part is divided into two steps: the first step generates the regression model, and the second step uses the neural network with the RNN2 approach.

## 2. Materials and Methods

### 2.1. Study Area

Reservoir areas are important for soil water conservation areas. Therefore, most reservoir areas are involved in many types of land use and other induced water quality issues. There are currently 95 reservoirs in Taiwan. The first stage of this study considered a total water area of about 700,000 $m^3$ of a reservoir, which is an appropriate size for analysis of water quality in a specific area. The effective capacity is 150,000 $m^3$, and the effective size is 716.8 $km^2$. Hence, the study selected Qingtan Weir (Figure 1) as an example to demonstrate the application on monitoring and prediction. The longitude and latitude of the Qingtan Weir are located at 24.948N and 121.545E, respectively. More specifically, this study took the Qingtan Weir water area sampling as a means to analyze monthly the past series of image data by a UAV, to monitor the water quality from images, and to produce the acquisition of UAV images. Then, the study built a new monitoring and prediction model based on an intelligent technology system, which is a computer system platform using a regression approach [8]. However, since one series of data was collected, the RNN model was applied to predict water quality in time series.

Due to the complicated components of the water body, this study decided to use the different changes in the color of the surface water of remote images to calculate the turbidity value of the water body through the reflectance of different wavebands [13]. Figure 1 displays the observation points, namely, A, B, and C, that were selected. The underwater sensor was mounted at those points, and time-series data were collected stably and automatically. In the first stage, we compared these three points and selected point B, which displayed the best outcomes of water quality. Due to points A and C being near to a sandy shore, some unpredictable values may have occurred as a result of rainfall. This study used multiple regression analysis methods to predict the turbidity value of water quality. The first trial used regression analysis. Then, it established the dependent variable (Y; turbidity) as a function of the independent variable (X; B, G, R, and IR) model and then estimated the parameters of the function based on the data obtained from the sample. The purpose of this was to present the image bands of the data to form a correlation of the turbidity of the water body.

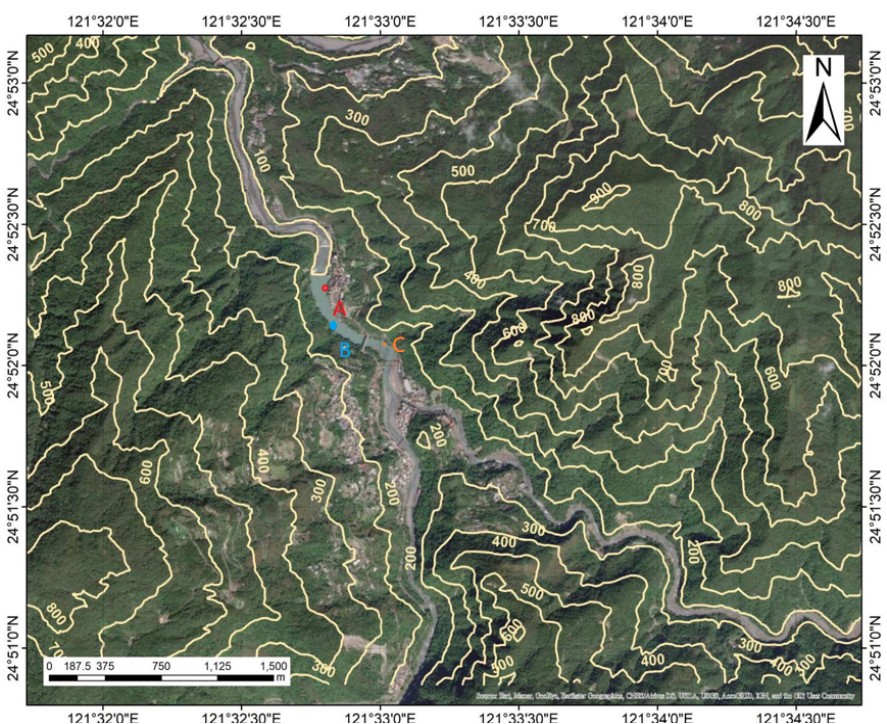

**Figure 1.** The contour elevation line of Qingtan Weir.

### 2.2. Study Plan

As previously mentioned, we received the image data from the UAV at point B in Figure 1. The water quality of turbidity was used as a series of outcomes to build the regression function by considering the image bands of B, G, R, and IR as input variables. The rainfall database was also used for calibration to amend the regression function. Figure 2 presents the research steps for analysis. The prediction of turbidity is quite difficult, especially considering the rainy hours of emergency notice for water treatment plants. The novelty this paper is that we used two neural network systems:

(a) A regular neural network to handle no-rain hours.
(b) The boosting model (RNN2) to handle rainy hours.

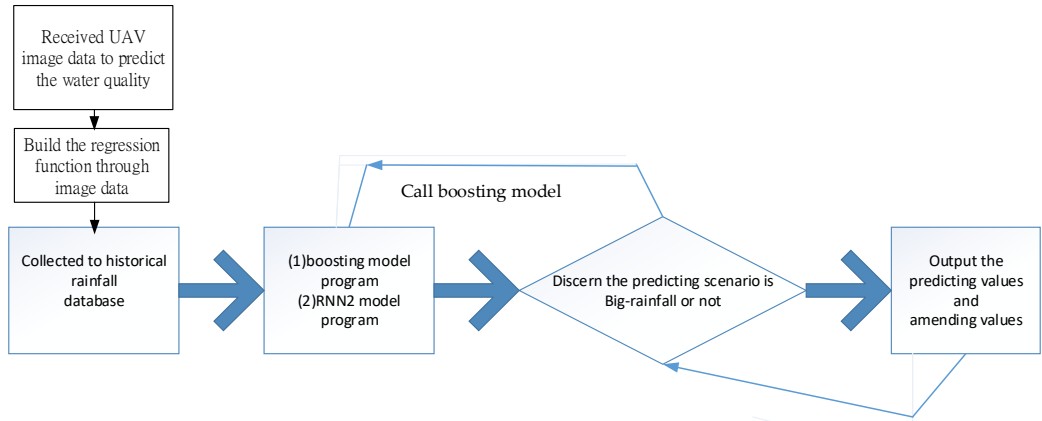

**Figure 2.** Steps for research analysis.

### 2.3. Image Data

The plan of the study was to use the acquisition image data at 10 a.m. of the selected study area in cloudless weather for a standard image format. Since the optical scanning operations are performed simultaneously, the image data will be operated to meet the

requirements of a scanning height of 15 m above the ground and FOV of 40° to obtain the ground resolution as 0.4 m images. At least 72 time intervals were recorded (locations are refer points A, B, and C in Figure 1) for the time-series data, which were used as an element to identify image features and make a prediction in future steps.

## 3. Methods

### 3.1. Neural Network (NN) Model

Neural networks are constructed based on the structure of the human brain, and they have contributed to current machine learning technology. They provide a simplified mathematical model to solve various nonlinear problems. In the past, researchers reviewed the ANN model's predictions of solar energy [14] and groundwater levels [15]. Past studies showed that the ANN model has a more accurate prediction than other conventional models, such as the Angstrom, conventional, linear, nonlinear, and fuzzy logic models [16].

The traditional neural network model comprise input, hidden, and output layers with auxiliary components, such as neurons, weight, bias, and activation functions [16]. Figure 3 shows basic neural network architecture with a multilayer perceptron. The input layers may receive input values from the signal or data, and the hidden layer analyzes the input values. The output layer collects the data from the hidden layer and decides the output. In the learning process, the neural network modifies its structure to obtain the same reference or set point as the supervisor. The training process is repeated until the difference between the neural network output and the supervisor lies within an acceptable range.

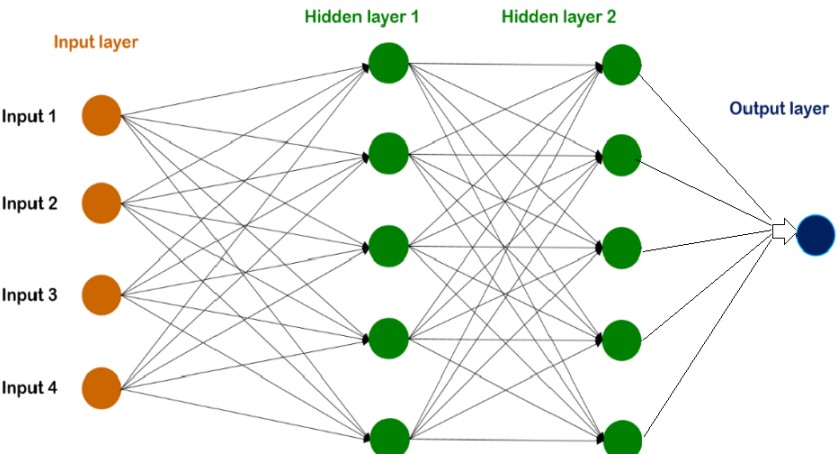

**Figure 3.** The traditional neural network model.

### 3.2. Recurrent Neural Network; RNN1

A recurrent neural network (RNN) is a class of artificial neural networks where connections between nodes form a directed graph along a temporal sequence. Previous RNN studies have drawn great attraction for water quality predictions [17–23]. This allows it to exhibit temporal dynamic behavior. Derived from feedforward neural networks, RNNs can use their internal state (memory) to process variable-length sequences of inputs. This makes them applicable to tasks such as unsegmented, connected handwriting recognition or speech recognition.

RNN models (see Figure 4) in particular are designed to analyze sequential data. They have been successfully used in fields such as speech recognition, machine translation, and image captioning. RNN processes sequence data by elements, and they preserve a state to represent the information at time steps. The traditional neural network assumes that all units of the input vectors are independent. Accordingly, the traditional neural network is ineffective for prediction when using sequential data. The architecture of RNNs comprise

three main components (input, hidden neuron, and activation function). The previous hidden state ($h_t$) can be formulated as

$$h_t = \tanh(U \cdot x_t + W \cdot h_{t-1})$$

where $x_t$ is the input at time $t$, $h_t$ is the hidden neuron at time $t$, U is the weight of the hidden layer, and W is the transition weights of the hidden layer. The input and previous hidden states are combined to produce information as the current and previous inputs go through a tanh function. Then, the output is the new hidden state, performing as the neural network memory because it holds information from the previous network. Training regular RNNs results in a series of vanishing and exploding gradient problems. In the case of the exploding gradient, the problem is resolved after backpropagation, which is closed at a certain point. However, the results may not be optimal because all the weights are not updated. In the case of the vanishing gradient, it can be adjusted by initializing the weights to reduce the possibility of a vanishing gradient [24].

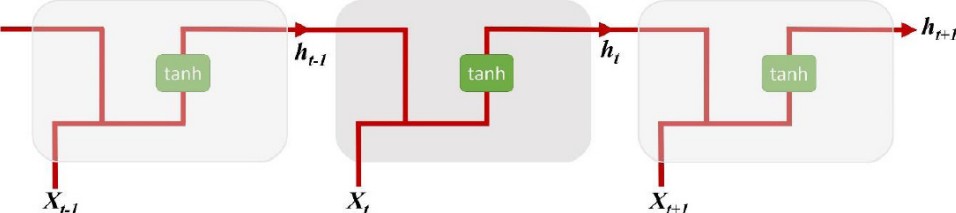

**Figure 4.** RNN model.

*3.3. Recursive Neural Network; RNN2*

A recursive neural network is a type of deep learning neural network. It is usually created by applying the same set of weights recursively over structured data for inputs. Then, it produces a structured prediction over variable-size input structures, or a scalar prediction on it, by traversing a given structure in topological order. Recursive neural networks have been successfully resolved of the prediction in learning sequence data [20–22]. A recursive neural network is a type of deep neural network produced by a set of weights recursively over a time series of structured inputs to produce consecutive predictions over variable-size input structures, or a scalar prediction on it [23–25]. It traverses a given structure in topological order. Recursive neural networks, sometimes abbreviated as RNN2 (see Figure 5), have been successful, for instance, in learning sequence and tree structures in natural language processing. RNN2 was first introduced to learn distributed representations of structure, such as logical terms.

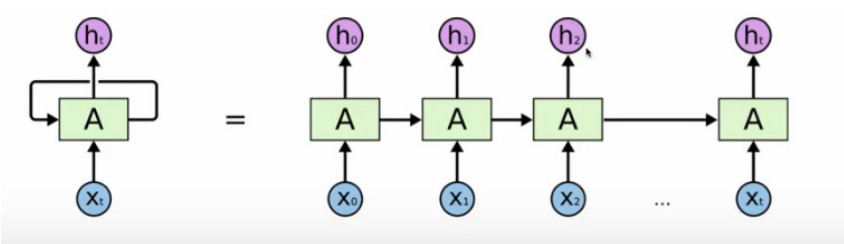

**Figure 5.** RNN2 model.

Although recursive neural network (RNN2) models demonstrate advantages in system robustness and time-series data computation, they have been widely applied to redundant data optimization. Existing RNN2 models suffer from local minimum and do not have planning completeness. Therefore, many techniques are used to solve the redundant data in time-series data acquisition [25,26]. Recursive neural networks usually operate on

any hierarchical structure, transforming child representations into parent representations. In general, stochastic gradient descent (SGD) is used to train this type of network. The gradient is calculated using backpropagation on varying data through a time series. In detail, every node in a given layer is connected to every other node in the next successive layer through a directed (one-way) connection. Each node (neuron) has a time-varying real-valued activation. Each connection (synapse) has a real-valued weight that can be modified. Nodes are either input nodes (receive data from outside the network), output nodes (produce results), or hidden nodes (modify data in the process from input to output). For supervised learning in discrete-time data, a sequence of real-valued input vectors arrives at the input node, one vector at a time. In any given time step, each non-input unit calculates its current activation (result) as a non-linear function of the weighted sum of the activations of all units connected to it [27,28]. The target activation given by the supervisor is applied to certain output units at a specific time step [29,30].

## 4. Discussion of the Results

Step 1: Build the regression function using image data.

In the first stage, the goal was to monitor turbidity in water. This study used multiple regression analysis to predict the turbidity value of water quality. B, G, R, and IR are four variables taken from image data, and the images were taken by a UAV. The turbidity values were extracted from sensor B. One series of the regression function was obtained. One of them can be formulated as

$$T = -1.238 \times R + 3.616 \times G - 2.991 \times B - 0.048 \times IR + 391.448, R^2 = 0.931 \quad (1)$$

where T is the turbidity.

Part of the regression error is plotted in Figure 6. The average error is about 0.07%. The Y-axis is the real value of turbidity, and the X-axis is the value predicted by regression.

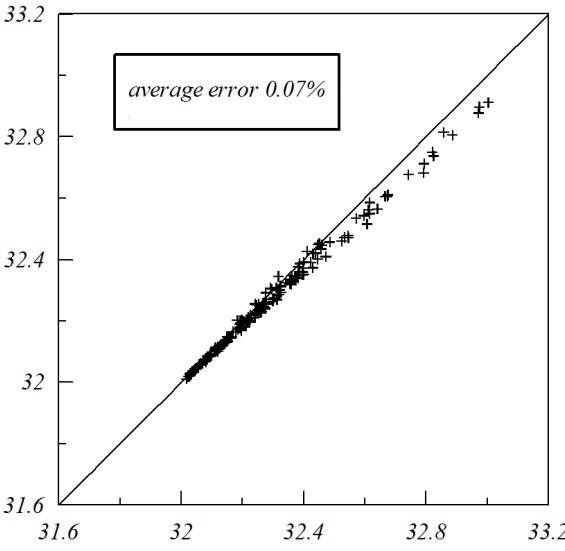

**Figure 6.** Prediction error of regression model ($R^2 = 0.931$).

Step 2: Use the time-series data of turbidity to build the RNN2 model.

The architecture of our RNN2 model was built based on a neural network framework, and it is shown in Figure 7. There is one input layer of neurons and one single output of neurons. Four hidden layers are constructed for computation. Two parts are constructed as (a) RNN2 parts and (b) NN parts. A total of 72 inputs of the neuron are set up to receive the signal of the water quality, considering the turbidity. Layer 2 is the hidden layer, and the received neurons of this layer take turns to transmit the data to the NN part of the system.

There are 72 input neurons of 12 h, with each hour taking 6 sampling rates. We selected four hidden layers, and the first two are in the RNN part and the last two are connected to the NN part. To accelerate and consider the convergence speed of the network, we decided to use the same number of neurons as in the input layer. The rectified linear unit (ReLu) was used as the activation function, which is used in the calculation of the core of the NN and RNN systems. Since the ReLu converts all the negative and zero input values to zero, using ReLu as an activation function for RNN2 is a legitimate concern. Srivastava et al. [31] suggested dropout to feedforward neural networks and RBMs and noted the probability of dropout being around 0.5 for hidden units and 0.2 for inputs, which worked well for a variety of tasks. There is a dropout ratio of 20%, which was set up in the first three layers. The last hidden layer was set up for 50% to accelerate the convergence of iterations.

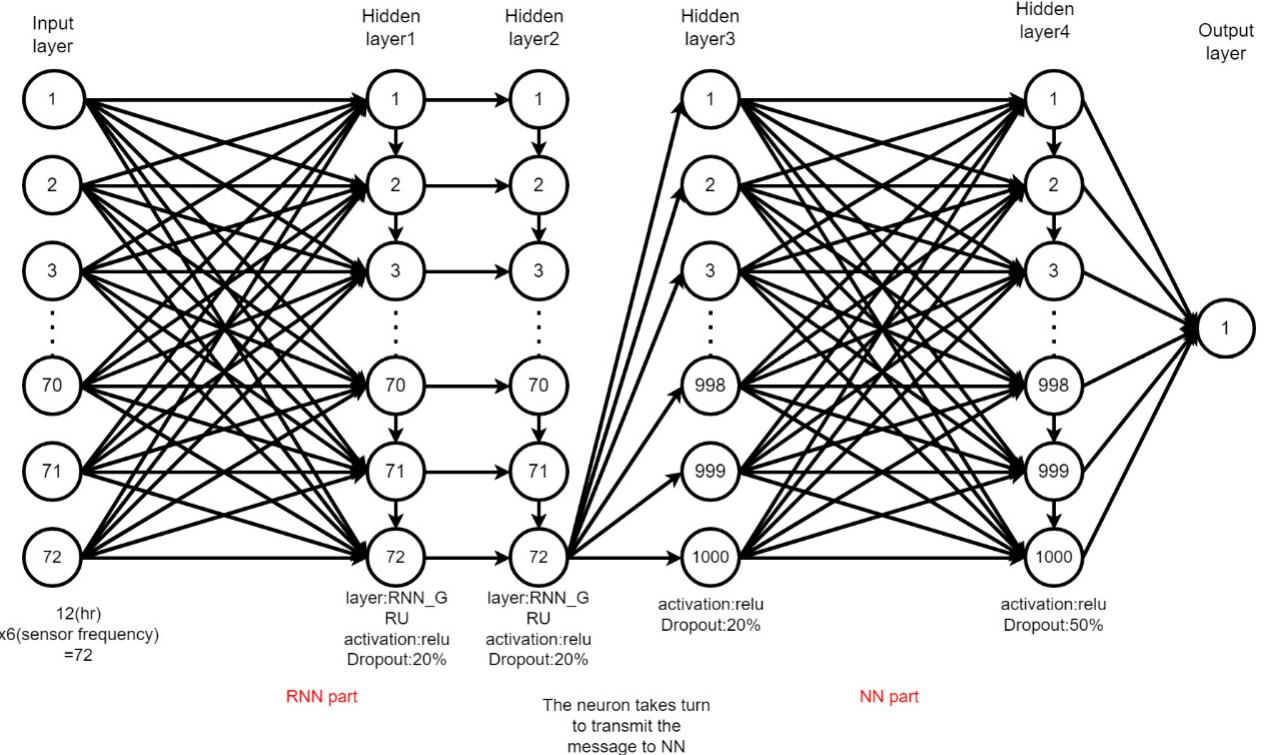

**Figure 7.** Flowchart of RNN2 design.

Step 3: The indicators of accuracy.
(1)  *RMSE*: Root-mean-squared error is a frequently used measure of the differences between values predicted by an estimator and the values observed. It can be formulated as

$$RMSE = \sqrt{\frac{\sum(y - \hat{y})^2}{T}}$$

where $T$ represents the different predictions, $\hat{y}$ is the value predicted by an estimator, and $y$ is the real value.
(2)  *MAE*: Mean absolute error (*MAE*) is a measurement of errors between paired observations with the absolute value.
(3)  $R^2$ is a statistical measure that represents the proportion of the variance for a dependent variable.
(4)  *CV*: Coefficient of variation is a statistical measure of the relative dispersion of data points in a data series around the mean. It can be formulated as

$$CV = \frac{\sigma}{\mu}$$

where $\sigma$ is the standard deviation, and $\mu$ is the mean.

### 4.1. Training Model

This study collected the training data in October 2019. The RNN2 model was built for two different cases of (a) accumulated rainfall corresponding to 0 and (b) accumulated rainfall greater than 0. The training dataset comprises 0.7 of the entire dataset, and the remainder is used as the testing dataset for the self-testing of the model. Figure 8 presents the prediction error distribution diagram.

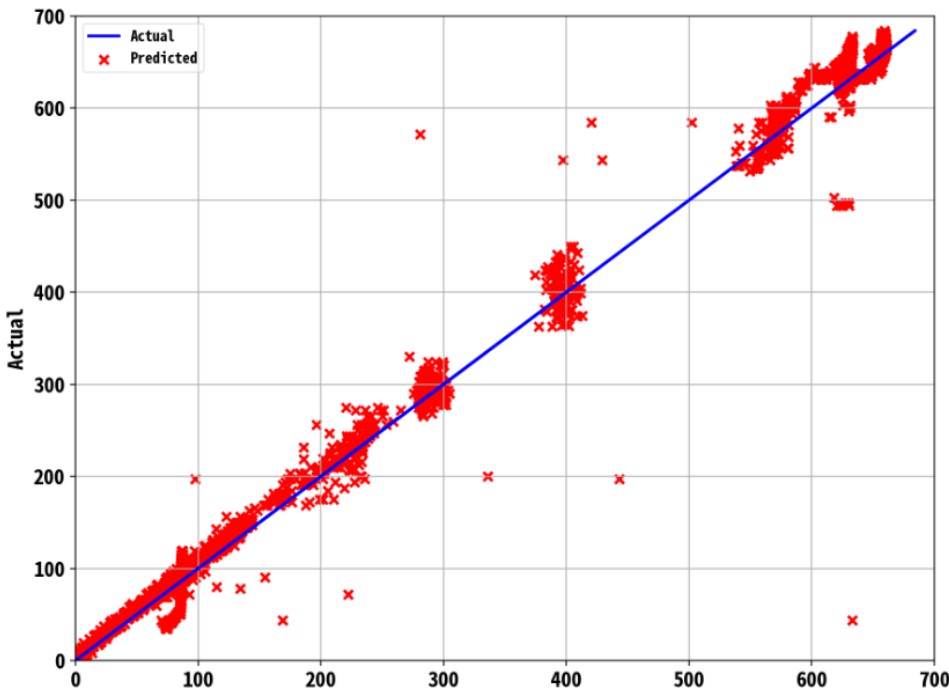

**Figure 8.** Prediction error of RNN2 ($R^2$ = 0.947).

### 4.2. Prediction Model for Convergence

RNN2 converges quickly, which is illustrated in Figure 9a for the case of no-rain days. It varies dramatically at the initial stage. The accuracy becomes stable at 16 epochs. The *RMSE* of the training dataset descends from 300,000. The test data move downward after the fifth epoch. Figure 9b shows the case of no-rain days of *RMSE* vs. epochs. The network converges at 14 epochs. The variations in Figure 9a,b are very similar.

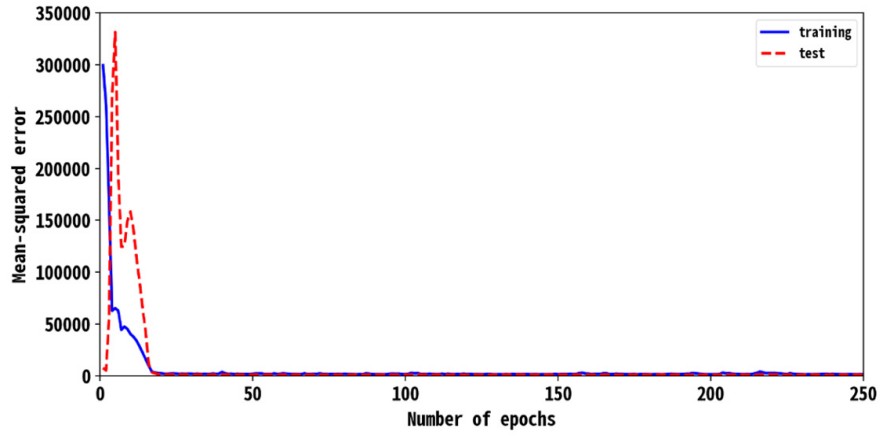

(**a**) Epochs vs. RMSE for no-rain days.

**Figure 9.** *Cont.*

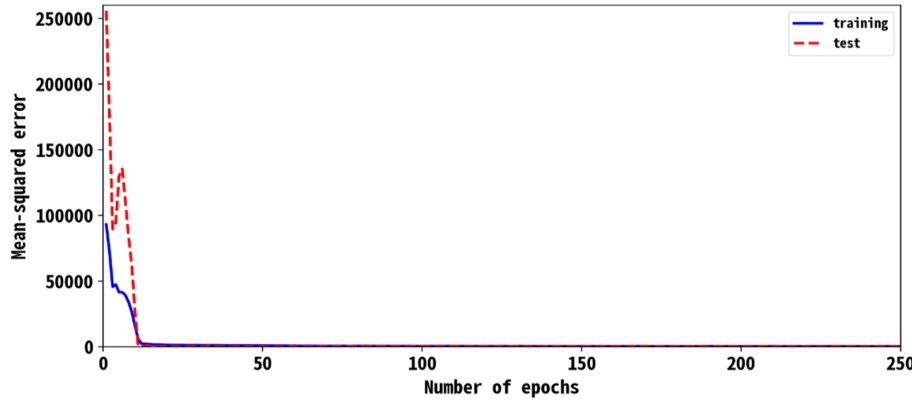

(**b**) Epochs vs. RMSE for rainy days.

**Figure 9.** Epochs of (**a**) no-rain days and (**b**) rainy days.

### 4.3. Prediction for Time–History

Time–history analysis is used to analyze the performance of RNN2 models. Accordingly, the turbidity of water can apply RNN2 for the step-wise analysis of temporal data. In our study, a portion of data is extracted to display the physical meaning of the analysis. The X-axis is the time and date, and the Y-axis is the density of turbidity. We used October 2018 to build the RNN2 model. Figure 10a shows the time–history of the accuracy of real data (blue) and verification (red) for October 2018. Figure 10b shows the time–history of accuracy of real data (blue) and prediction (red) for November 2018. The *RMSE* of October is 20.89, which is smaller than that of November (36.11); this makes sense as the prediction outcomes are greater. However, the differences between October and November are sufficiently small to explain the prediction of outcomes, which are considered quite rational.

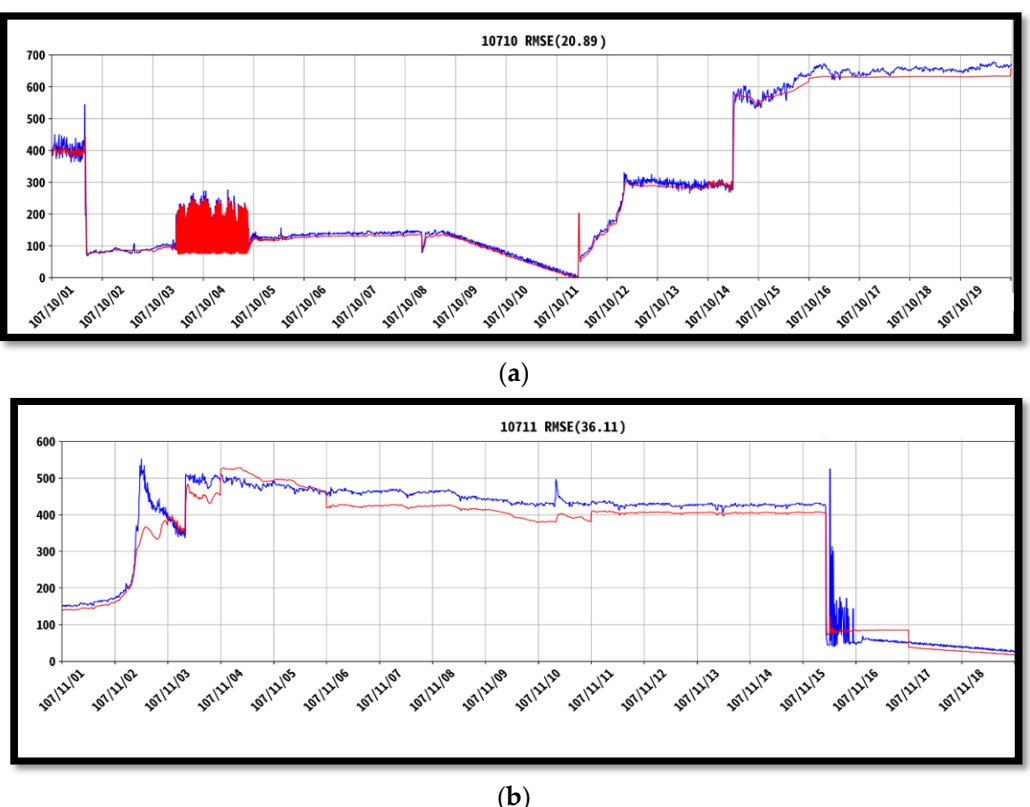

**Figure 10.** Time–history difference between real (blue) and predicted (red). (**a**) Time–history of October for building of RNN2 model; (**b**) time–history of November for predicting of RNN2 model.

In addition to using time–history analysis, we also employed MAE, *RMSE*, *CV*, and $R^2$ to observe the performance and accuracy of the model. The performance criteria of RNN2 are presented in Table 1. We illustrated the *MAE*, *RMSE*, and $R^2$ of October 2018 and November 2018. October represents the training data used to generate the RNN2 model. November represents the verification part, using testing data to examine how the real-time value is displayed.

**Table 1.** The RNN performance criteria table.

| Date | Type | Value |
|---|---|---|
| October 2018 | *MAE* | 13.88 |
| October 2018 | *RMSE* | 20.89 |
| October 2018 | *CV* | 0.757 |
| October 2018 | $R^2$ | 0.993 |
| November 2018 | *MAE* | 25.20 |
| November 2018 | *RMSE* | 36.11 |
| November 2018 | *CV* | 0.768 |
| November 2018 | $R^2$ | 0.961 |

## 5. Summary and Conclusions

The recursive neural network (RNN2) is a type of artificial neural network that uses time-series data to predict a few forward step-wise data items of values. Such deep learning algorithms are usually designed for temporal problems. In addition, RNN2 utilizes training data to learn a learning system, which is similar to its "memory", as it takes information from prior inputs to influence current outputs. Traditional deep neural networks presume that inputs and outputs are independent of each other, as well as the output of recurrent neural networks depending on prior elements within a sequence. When future events help to determine the output of a given sequence, unidirectional recurrent neural networks cannot take account of these events in their predictions. Considering these characteristics of RNN2, the few-step time-series data of water components can be successfully applied by using RNN2 to resolve them.

Turbidity is the most important measure indicator of the relative clarity of water. It is an optical feature of water, and it is a measurement of the amount of light that is scattered by materials in water samples. In general, the higher the intensity of scattered light, the higher the turbidity. However, many materials, such as clay, silt, very tiny inorganic and organic matter, algae, dissolved organic/non-organic compounds, plankton, and other microscopic organisms, cause water to be turbid. We monitored a reservoir's turbidity values using sensors mounted at the waterbed. Simultaneously, a series of images from the UAV were taken to correlate turbidity and the band values of B, G, R, and IR. Remote sensing data were used to present the values of turbidity of water in our selected reservoir. The R-squared value is 0.931, which satisfies the prediction of a regression model. We aimed to construct an ANN system (regular no-rain hours) with an RNN2 system for a subroutine (considering rainy hours) to predict turbidity values using water sensors. Due to the occurrence of rainy hours, the prediction can have huge errors, but our subroutine can adjust the accuracy of results. As part of this study, a deep learning approach using the RNN2 model is presented to forecast time-series data by operating the main program in regular hours. That is, when the data increase dramatically, the subroutine is automatically operated. The main objectives of this work were to design a model that can not only predict the very next step but also generate a sequence of predictions and utilize multiple driving time series together with a set of static (scalar) features as its inputs. The *RMSE* of October is 20.89, which is smaller than that of November (36.11). The predicted R-squared values in RNN2 are 0.993 and 0.941. The differences in *RMSE* for October and November are sufficiently small, which explains why the predictions of outcomes are considered quite rational.

**Author Contributions:** Conceptualization, S.W.; programing, H.-L.M.; formal analysis, M.-L.Y.; writing—original draft preparation, S.W.; writing—review and editing, T.-Y.C.; project administration, M.-L.Y. All authors have read and agreed to the published version of the manuscript.

**Funding:** The APC was funded by Ministry of Science and Technology.

**Institutional Review Board Statement:** Not applicable.

**Informed Consent Statement:** Not applicable.

**Data Availability Statement:** The data presented in this study are available on request from the corresponding author.

**Acknowledgments:** The authors express their gratitude to the National Science Council (MOST 109-2622-M-275-001 -CC3) for sponsoring this work.

**Conflicts of Interest:** The authors declare no conflict of interest.

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
