# Peer review of "The Robust Study of Deep Learning Recursive Neural Network for Predicting of Turbidity of Water"

_water, doi:10.3390/w14050761_

Round 1

Reviewer 1 Report

To be candid i did not find this paper suitable for journal like water(mdpi) for several reasons. 

There is a single novelty in this study to board team of world scientist. The paper title did not reflect the main text in this work. the authors try to predict turbidity and all the explanation is on water quality note that water quality is very broad area comprised of hydrochemical, physicochemical and etc.  

Authors mix methodology in the result section, the discussion of the paper is very weak. i did not see a rich introduction and motivation of the study and backup recently literature.

All the figures in this paper are not of good quality nor the standard of scientific paper like water mdpi.

The reference is wrong and the style of the paper as well as the arrangement all not in order.

The captions of all the figures are wrong

There is no performance criteria table how do you get the results then

The RMSE alone can not indicator to assess the models several indicators need to be introduced how can we agree with this work

Prediction of turbidity has been developed using several deep learning, hybrid, ensemble and optimization what is your contribution

Author Response

Thank you for you spending a lot of time giving us suggestions. We do our best to modify the paper. Please see the attached file.

Reviewer 2 Report

The paper is based on an interesting idea but has significant flaws, which need to be fixed. Please consider my following comments:

- The literature review is poor.

- The knowledge gap shown by previous similar literature studies should be clearly delineated in the introduction. Furthermore, in the last paragraph of the introductory section, the authors should clearly highlight the objectives of this study and the innovative contribution of this study in overcoming the limitations of existing studies in the literature.

- The parameters of the models must be indicated in detail and clearly.

- A proper discussion of the results is missing. The results must be discussed in a separate section and contextualized with appropriate comparisons with the existing literature.

- Authors are advised to also consider additional metrics for the evaluation of results and additional graphical representations (e.g. radar plots, box plots).

- To better address the previous comments, and to improve the overall quality of the article, authors may consider the following additional references:

Barzegar, R., Aalami, M. T., & Adamowski, J. (2020). Short-term water quality variable prediction using a hybrid CNN–LSTM deep learning model. Stochastic Environmental Research and Risk Assessment, 1-19.

Ferreira, L. B., & da Cunha, F. F. (2020). Multi-step ahead forecasting of daily reference evapotranspiration using deep learning. Computers and Electronics in Agriculture178, 105728.

Granata, F., & Di Nunno, F. (2021). Forecasting evapotranspiration in different climates using ensembles of recurrent neural networks. Agricultural Water Management255, 107040.

- The quality of the figures needs to be improved.

- English must be reviewed by a native speaker.

Author Response

We deeply appreciate your help on reading our manuscript.

Also the entire is proof-reading by an English editor to fix the grammatical errors.

please see the attached file

Reviewer 3 Report

The manuscript handles with the study of monitoring water quality systems, although contains some questions.

In my opinion, the abstract has to be reformulated. It's quite extensive, mainly in the references for the background of the study, and on other hand don’t make adequate reference to the conclusions of the study.

Line 41-46: these part the way it is written is material and methods and not the introduction section.

The first two paragraphs of the intro are very confusing, almost looking like a section of the discussion of the results and not the introduction.

The prediction model is interesting, but then what is its use, for example, in drinking water treatment plants. It seems to me that applicability, or reference to it, is lacking. Because in the end, it is achieved a measurement of turbidity when it rains or not.

Author Response

We follow your instructions to amend the paper. We also express our gratitude for your suggestions and advice.

Please the attached file.

Round 2

Reviewer 1 Report

  1. the title is technically wrong please write this 

     The Robust Study on Deep Learning Recursive Neural Network for Predicting the Turbidity of Water

  2. Extensive English require
  3. what is... ds of B, G, R, IR to build the.... in the abstract
  4. What is RNN2 in the abstract
  5. ....  two parts: (a)The first ... is wong correct it please  also .... and (b)The second 
  6. Table heading .. data... should be standard for the rest of the heading 
  7. Provide the formulas used in Table or separate section for performance criteria

Author Response

Thank you for your advice.

We follow your suggestions! please see attached file.

Reviewer 2 Report

The authors made a good effort to address my comments and improve the article. I have no further comments.

Author Response

Thank you very much.

We carefully review the English Manuscript and amend the minor errors.